# Total Carbon Content Assessed by UAS Near-Infrared Imagery as a New Fire Severity Metric

Anna Brook [1,*], Seham Hamzi [2], Dar Roberts [3], Charles Ichoku [4], Nurit Shtober-Zisu [5] and Lea Wittenberg [6]

1 Spectroscopy & Remote Sensing Laboratory, Spatial Analysis Research Center (UHCSISR), Department of Geography and Environmental Studies, University of Haifa, Mount Carmel, Haifa 3498838, Israel
2 College of Sakhnin for Teacher Education, Sakhnin 20173, Israel; seham_l@sakhnin.ac.il
3 Geography Department, University of California Santa Barbara, Santa Barbara, CA 93106, USA; dar@geog.ucsb.edu
4 NASA Goddard Space Flight Center, Greenbelt, ML 20771, USA; charles.m.ichoku@nasa.gov
5 Department of Israel Studies, University of Haifa, Haifa 3498838, Israel; nshtober@research.haifa.ac.il
6 Geomorphology Laboratory, Department of Geography and Environmental Studies, University of Haifa, Haifa 3498838, Israel; leaw@geo.haifa.ac.il
* Correspondence: abrook@geo.haifa.ac.il

**Abstract:** The ash produced by forest fires is a complex mixture of organic and inorganic particles with many properties. Amounts of ash and char are used to roughly evaluate the impacts of a fire on nutrient cycling and ecosystem recovery. Numerous studies have suggested that fire severity can be assessed by measuring changes in ash characteristics. Traditional methods to determine fire severity are based on in situ observations, and visual approximation of changes in the forest floor and soil which are both laborious and subjective. These measures primarily reflect the level of consumption of organic layers, the deposition of ash, particularly its depth and color, and fire-induced changes in the soil. Recent studies suggested adding remote sensing techniques to the field observations and using machine learning and spectral indices to assess the effects of fires on ecosystems. While index thresholding can be easily implemented, its effectiveness over large areas is limited to pattern coverage of forest type and fire regimes. Machine learning algorithms, on the other hand, allow multivariate classifications, but learning is complex and time-consuming when analyzing space-time series. Therefore, there is currently no consensus regarding a quantitative index of fire severity. Considering that wildfires play a major role in controlling forest carbon storage and cycling in fire-suppressed forests, this study examines the use of low-cost multispectral imagery across visible and near-infrared regions collected by unmanned aerial systems to determine fire severity according to the color and chemical properties of vegetation ash. The use of multispectral imagery data might reduce the lack of precision that is part of manual color matching and produce a vast and accurate spatio-temporal severity map. The suggested severity map is based on spectral information used to evaluate chemical/mineralogical changes by deep learning algorithms. These methods quantify total carbon content and assess the corresponding fire intensity that is required to form a particular residue. By designing three learning algorithms (PLS-DA, ANN, and 1-D CNN) for two datasets (RGB images and Munsell color versus Unmanned Aerial System (UAS)-based multispectral imagery) the multispectral prediction results were excellent. Therefore, deep network-based near-infrared remote sensing technology has the potential to become an alternative reliable method to assess fire severity.

**Keywords:** fire severity; post-fire environment; total carbon; spectral model; machine learning; unmanned aerial system multispectral imagery

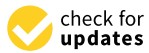



## 1. Introduction

Ash and char production can be used as broad indicators of the temperature reached or heat produced during a fire and can used to understand the impacts of the fire on nutrient cycling and landscape recovery [1]. Recent studies have investigated changes in

ash characteristics, such as color, as an indicator of fire severity [2–4]. Ash color can reflect the severity of fire in the immediate area and can range from heterogeneous black ash (char) produced in low-intensity fires to fine, homogenous white-grey ash produced during higher intensity fires [5,6]. Fire severity is an indirect measure of how fire intensity impacts ecosystems assessed by organic matter loss [7]. Low-to-medium-severity fires may have positive effects on soils, such as increased nutrient availability [2]. The negative impacts of severe fires include a reduction in the organic content of the soil [8], modification of soil structure [9–11], mineralogical changes [12], and deleterious changes to hydrological responses [13].

The most widely used method for subjectively determining soil color is by comparison of samples with the Munsell soil-color chart [14]. The Munsell soil-color chart does not allow numerical or statistical analysis as the color space is divided into non-contiguous representation pages. While ash color is an excellent indicator of complete combustion, color can be perceived and described differently, even when matching against a standard chart, and color assessment is affected by ambient light.

Vibrational spectroscopy deals with the visible and near-infrared (VNIR) and mid-infrared (MIR) region of the electromagnetic spectrum and includes a range of techniques involving the absorption of electromagnetic energy based on the vibrational modes of molecules. Fundamentals of these vibrational absorption bands appear in the MIR region. They are strong, distinct, and can act like fingerprints, i.e., be used to identify specific chemical bonds associated with the bands. Overtones and combinations of fundamental bands characterize the VNIR region. They are generally weaker, overlapped, and challenging to resolve for specific chemical constituents. VNIR reflectance regions are rapid, require small samples, and are very reproducible, and these advantages have made VNIR spectroscopy a powerful tool in soil organic matter (SOM) chemistry. Moreover, the VNIR region illustrates combinations and overtones of vibrational frequencies of bonds such as those between oxygen and hydrogen (OH), carbon and hydrogen (CH), as well as nitrogen and hydrogen (NH) producing indicative NIR absorbance bands [15].

NIR can be used to describe the residual matter following combustion and may be able to expose the mineralogy of samples of char and ash [16]. Therefore, we may be able to use the spectral data from the NIR region collected from ash to estimate changes in the levels of carbon (C), nitrogen (N), and other nutrients [17]. The combinations and overtone features across the NIR region correspond to a fundamental absorption across the MIR region measured by diffuse reflectance infrared Fourier Transform (DRIFT) and transmission mode. These measurement modes differ in resolution and sensitivity; whereas DRIFT is better at revealing structural information about organic matter, the transmission spectra are better regarding inorganic material content, but cannot be applied directly to samples. Many studies describe methods of applying infrared techniques to soil organic matter [18], usually by direct application to a soil sample and studying relatively simple mineral-organic matter systems. Studies have also examined the potential of diffuse reflectance mid- and near-infrared spectroscopy [19] to determine not only carbon content but also levels of metals such as Co, Cr, Ni.

C levels represent the continuum of organic materials transformed by different degrees of burning and can be found in many post-fire patches [20,21] with nitrogen and other elements [5,22,23]. Incomplete combustion in wildfires produces Total C (TC), where some of the biomass is transformed into pyrogenic organic matter, e.g., charcoal and black carbon [24]. In general, TC is a part of soil organic carbon (SOC) which is a major planetary resource supporting ecosystem services and the realization of some of the 17 Sustainable Development Goals (SDGs).

The low correlation between C and pyrogenic carbon (PyC) gains or losses and fire severity, reported by [25], highlights the complex impacts of fire on forest C. The severity of fire indicates the magnitude of its impact on an ecosystem and is described using numerous definitions and metrics [7,26]. The correlations between fire severity and TC budgets are complex, because wildfires release C, and also produce PyC that contributes

to C sequestration in soils [27,28]. The chemistry of these molecules is heterogeneous and can be considered as a mixture of compounds ranging from the slightly charred biomass to the highly condensed aromatic materials [29,30] with an overall increase in TC concentration (%).

One study [6] provided the first estimation of pre- and post-fire PyC stocks in a boreal jack pine forest by calculating changes in C concentration in a high-severity fire neglecting impacts on mineral soil. A more recent study [25] on fire effects on forest C and PyC in a mixed-conifer forest using pre-and post-fire measurements following five wildfire events concluded that using post-fire severity estimates does not give a complete picture. The limitations and challenges in extrapolating fire severity according to C concentrations at broader scales are clear.

Remote sensing (RS) tools are widely used for broad-scale fire severity estimation and mapping via multi-date change detection [31,32]. The standard RS-based indices include the Normalized Burn Ratio (NBR), differenced Normalized Burn Ratio (dNBR), soil-adjusted vegetation index SAVI [33,34], and burned area index BAI [35]. Additionally, field classification of composite effects has been developed and applied [36,37].

Bi-temporal vegetation indices (e.g., differenced NDVI) are used to create a continuous differenced raster to quantify an absolute measure of change. Continuous differenced index values are used to characterize fire severity by field survey or aerial photo interpretation, where fire severity is defined as the loss of biomass [7]. All indices are 60–70% accurate compared to field validation [38,39] of the spatial variations in severity within a single fire. Since all suggested indices are sensitive to post-fire changes, they are generally accepted as robust methods for determining fire scars in the landscape and assessing the recovery of the vegetation [40–42].

Nevertheless, categorizing the values of spectral indices into standardized (consistent between fires and landscapes) severity groups is complex and almost impossible. Furthermore, the fact that fire severity indices distinguish between photosynthetic and non-photosynthetic ground targets means that the pre-fire conditions such as vegetation structure, moisture, soil type, and topography [43,44] were not considered.

Given that all the state-of-the-art spectral indices for fire severity were developed for space-borne remote sensors, their main disadvantages are their reliance on physical landscape parameters, satellite data quality (spatial, radiometric, and spectral resolutions), and availability (temporal resolution). Thus, many methods are compromised because of the available imagery, spatial, spectral, and temporal resolution, and access to ground sampling data [45]. Moreover, the accuracy of fire severity mapping depends on location, and the target ecosystem, and not only on sensor specifications [46]. Considering that wildland areas are typically characterized by dense canopies and high topographic relief with many topographic shadows, they are very challenging for space-borne remote sensors to capture. Furthermore, the coarse spatial resolution and operational limitations of broad-scale mapping, yield practically useless products for mapping the burnt understory or low fire severity [47,48].

Advances in sensor technology provide new opportunities for fire severity mapping and alternatives to broadband, low to medium resolution multispectral satellite sensors. Hyperspectral airborne sensors, e.g., AVIRIS, have been widely used to map fire severity in the USA [32,49]. Similarly, studies have shown the benefits of using active sensors such as light detection and ranging (LIDAR) [50], Radio Detection and Ranging (RADAR) [51], and Terrestrial Laser Scanning [52]. The developments in Unmanned Aerial System (UAS) technology also advanced sensor performance and miniaturization [53]. These systems are capable of carrying multispectral, high spatial resolution spectrometers, LiDAR, and thermal sensors. UAS products have the potential to determine initial and extended fire impacts and offer options for high spatial and temporal severity assessments of burnt sites [53]. The thermal sensors on UAS have been used to map actively burning fires [54], and rarely to assess vegetation burn severity [55]. Yet, a post-fire digital terrain model

(DTM) derived from a UAS survey was differenced from a pre-fire DTM from airborne LiDAR, to estimate the depth of surface burning in a tropical peatland [56].

These "big data", i.e., the multi-sensor spectral data cannot be analyzed using conventional methods, e.g., multiple linear regression or partial least squares (PLS) regression models. Partial least squares-discriminant analysis (PLS-DA) is a supervised modelling method that uses a PLS algorithm to predict the membership of a sample or spectrum to a given class. It is often used to deal with the multicollinearity problem in near-infrared (NIR) spectra because of the very high inter-correlation between measured absorbance at sequential wavelengths in spectral data analysis.

The current study aims to develop a novel method to assess fire severity using UAS-based remote sensing. The main goal is to determine whether the UAS multispectral (very high spatial resolution) data across visible and near-infrared (VNIR) regions, collected from post-fire environments and advanced classification models, can be used as a fire severity metric. The suggested supervised and unsupervised shallow and deep classification models are: PLS-DA, ANN, and 1-D CNN. The main contribution in this work is a predictive model for the content of total carbon (TC) in residues. Our TC model will have practical applications because of the findings that in post-wildfire environments, TC content increases correlate with field-based fire severity assessments [25].

## 2. Materials and Methods

### 2.1. Study Area and Sample Collection

Mt. Carmel (32.7699°N/35.0657°E), a mountain range in Northwestern Israel, has been subjected to an increasing number of wildfires of various extents and severities. The region represents a typical Mediterranean ecosystem with relatively long, hot, dry summers and short (4–5 months) rainy winters. Spring and autumn months are often characterized by considerable variations in temperature relative humidity and precipitation. 'Sharav'/heat wave episodes, with strong hot and dry eastern winds, are common in the transition seasons, increasing the risk of extensive wildfires. The experimental datasets derived from three sources are summarized in Table 1.

Before collecting ground samples in both experimental and urban wildfire sites, the spectral data were measured with portable field spectrometers OceanOptics USB4000. The wavelength-dependent signal-to-noise ratio (S/N) for our instrument was estimated by taking repeat irradiance measurements of the Spectralon® (Labsphere Inc., North Sutton, NH, USA) white reference panel over a 10-min period and analyzing the spectral variation over this time. The value for each ground sample was calculated by averaging 3 spectra of both radiance and reflectance measurements during the UAS campaign. The reflectance was calibrated against a Spectralon® white reference panel. The optimization procedure was programmed to work in both radiance and reflectance modes, averaging 10 replications per measured spectrum. Each ground target (30 × 30 cm) was measured systematically by collecting about 10 points, and the targets were uniformly distributed across the study site, creating a grid/matrix over the site. All points in a designed matrix were about 5 m apart and the spectral measurement was taken from 1 m height with a bare-optics of a 24° field of view (about 60 cm$^2$ footprint on the ground) with spectral error (standard deviation).

The imagery data were collected using two sensors on the UAS: RGB camera, and RedEdge-MX Micasense camera (5 data bands) with output bit depth of 12 bit simultaneously with spectral measurements. The DJI Phantom 4 Pro RGB images with a pixel size of 1.2 cm at ~60 m flight altitude. The multispectral data were acquired by Matrice 600 Pro DJI at ~30 m flight altitude with a pixel size of 2 cm. The cameras were stabilized in pitch, roll, and yaw by a three-axis gimbal and followed pre-programmed flight plans to assure complete coverage of a frontal overlap of 90% and an adjusted side overlap (by a number of flights) using an autopilot module of Pix4D capture application. The image sequences were collected in perpendicular flight lines using the autopilot 'double grid' software option. The absolute vertical and horizontal accuracies were improved using the dGPS system in

the field. To that end, 12 ground control points, evenly distributed throughout the study area, and near-visible horizontally and vertically important objects were measured.

**Table 1.** Datasets.

| | Laboratory Dataset | Controlled Field Experiment Dataset | Urban Wildfire Dataset |
|---|---|---|---|
| Location | 3 unburnt plots (20 m$^2$). Location Mt. Carmel (32°43′16.3″N 35°00′15.8″E). | Isolated 2 × 2 m area burned by an open fire without interference and without combustion accelerators. Location near the University of Haifa (32°45′28.0″N 35°01′27.0″E). | Site size 550 × 150 m; Before the fire, more than 70% of the site was covered by vegetation and more than 50% of the vegetation was trees. Location Haifa (32°46′54.3″N 34°59′56.7″E). |
| Event Description | | The experiment took place in July 2017. Air temperature 26 °C, average wind speed 2 m/s, soil temperature 46 °C, litter temperature 38 °C. | In November 2016 following a typically hot dry summer and unusually dry autumn, a wave of fires hit Israel. There were more than 170 wildfire events. The fire suppression activities in Haifa took nearly 24 h. The total burned area was 13 ha. |
| Sample Collection | Samples were collected using a circular sampling ring and leaves, twigs, soil, and fine fuel were placed in separate bags. | Multiple subsamples at evenly spaced intervals along a transect radiating from a centroid were collected and composited. | Samples were collected on November 26th from an almost fully burned site. The top-ash samples (at a depth of 1–3 cm) were collected along a transect radiating from a centroid at the site. |
| Sample Description | The vegetation is broadly classified as a Mediterranean forest and the predominant species are *P. halepensis* and *P. lentiscus*. OM1 is herbaceous (n = 50) OM2 is a mixed sample of leaves and twigs of *P. lentiscus*, *C. salviifolius*, and herbaceous vegetation at a size of approximately 5–7 cm (n = 50) OM3 is the needles of *P. halepensis* (n = 50) OM4 is the leaves of *P. lentiscus* (n = 50) OM5 is the twigs of *P. halepensis* (n = 50) OM6 is the twigs of *P. lentiscus* (n = 50). | The vegetation is mainly composed of annual herbaceous species partially covered by the needles and branches of *P. halepensis*. Note that the summer months are very dry. | The natural vegetation is composed of Pinus halepensis, *Quercus* spp. and *Pistacia* spp. *Pinus halepensis* and *Quercus* spp. have relatively short time-to-ignition and long flame duration, relegating them to the class of extremely flammable vegetation. |

### 2.2. Sample Treatments and Measurements

The laboratory soil samples were oven-dried for 48 h at 60 °C and then sieved (<2 mm) to remove stones, plant debris, and large root matter. In total 300 soil samples were prepared in 500 mL beakers; each sample had a 30 g 13C-depleted biomass placed on top of 80 g of soil. The samples (see Table 1) were heated in a muffle furnace for 2 h [3] at 250 to 600 °C at intervals of 50 °C. After cooling to room temperature overnight, all samples were scanned with the proposed sensors: portable field spectrometers OceanOptics USB4000 using a deuterium-tungsten halogen light source (a single lamp with a wide spectral output, covering the entire UV-Visible-NIR range) and a bifurcated optical fiber with two fibers in the common end, DJI Phantom 4 Pro RGB camera, and RedEdge-MX Micasense camera. The samples were then subdivided into laminas following textural and color variations [16,56]. Only the top ash layer was investigated; it should be noted that the combustion of plant

biomass in a muffle furnace is not identical to field burning [57,58]. The flaming combustion creates diverse heat waves over time and results in mixed residues/burned matter.

All the collected top-ash samples (laboratory and field) were spectrally measured across the mid-infrared region (MIR) to quantify total C content [59] by applying diffuse reflectance Fourier transforms infrared (DR-FTIR) spectroscopy. 10 mg of the isolated ash layer was mixed with 200 mg of spectroscopic-grade KBr (Aldrich Chemical Co. Milwaukee, Brookfield, WI, USA) for analysis. The mixture was initially hand-ground and then ground again in a Wig-L-bug using a stainless steel vial with a stainless steel ball pestle for 45 s. FTIR measurements were performed with a Bruker TENSOR series FTIR spectrophotometer (Bruker, Ettlingen, Germany) equipped with a Pike EasiDiff optical bench. The scans were from 4000 to 400 $cm^{-1}$ range with a resolution of 4 $cm^{-1}$. The spectrum of each sample was normalized against the background spectrum obtained from the diffuse reflectance spectrum of pure KBr. Spectra were collected as reflectance and Kubelka-Munk units, they were not smoothed, and the original resolution was kept for further spectral analyses by the OPUS software. Moreover, a set of 24 selected samples were analyzed using a Leco TruSpec CHN to determine and confirm the total carbon (TC) content.

### 2.3. Spectral and Imagery Data Pre-Processing

The spectral calibration process was normalized using measurements of an internal standard. This process enables the isolation of noisy wavelengths (from the signal) and the generation of a noise-less (smooth) data set for further analysis. The reflectance spectra of the internal standard and the measured reflectance spectra were normalized to the continuum level by interactively choosing continuum points, linearly interpolating between them, and then dividing the final spectrum by the continuum. The measure of internal error mostly reflects the operator's consistency in choosing the continuum level. A normalization factor is obtained from the calculated ratio between continuum-removed spectra (spectral repetitions) of the internal standard [60].

Imagery pre-processing was carried out on *RGB* and multispectral UAS-based data [61]. The *RGB* color information of each pixel was calibrated and normalized against *Spectralon* reflectance using Equation (1). The normalization step is performed to remove the undesirable effect of noisy pixels.

$$RGB_{norm} = 2^n RGB_{target} / RGB_{spectralon} \tag{1}$$

where $n$ is the number of bits per pixel for each color band.

The images were processed via the structure from motion (SfM) method, implemented in Pix4D Mapper Pro (v. 4.3.31, Pix4D, Prilly, Switzerland), which completes all the main SfM steps. The SfM workflow starts with feature identification, followed by feature matching, camera model optimization, and the final step is the bundle block adjustment [62]. The point density option was set to 'optimal' and the minimum number of matches was set to three using a matching window size of $9 \times 9$ pixels. The accuracy of RGB based digital surface model (DSM) is 2 cm in the horizontal (X, Y) coordinates, and <3 cm in the vertical (Z) coordinate, calculated for 4 validation GPS points that were measured in the field but were not used in the Pix4D model. The multispectral imagery was not processed by bundle block adjustment, to preserve the radiometry collected by the system. The orthophoto produced in Pix4D Mapper for RGB images was converted to the Munsell color system using the methodology published by [63].

The multispectral UAS-based orthophoto is a radiometric corrected by an advanced supervised vicarious calibration (SVC) method [60] and the bidirectional reflectance distribution function (BRDF) applies correction coefficients to every pixel in a point-cloud and calculates the at-sensor radiance for each pixel according to Figure 1. A depression angle is calculated for each point in the cloud. This angle represents the orientation of a given point/pixel towards the sensor. Once the point/pixel/surface is facing the sensor (in nadir), the calculated angle equals 90°. The angle describes the situation that when the point/pixel/surface is tilted then its radiance is scattered and reflected in an off-nadir

way. The calculated ground target depression angle is used to correct solar information (azimuth and zenith), which is calculated at a given date, time, and geographic location at a central coordinate. The corresponding solar information is used to retrieve the BRDF correction coefficients for the SVC calibration nets target [64]. The calculated coefficients are then applied to the entire scanned scene, the full point cloud data.

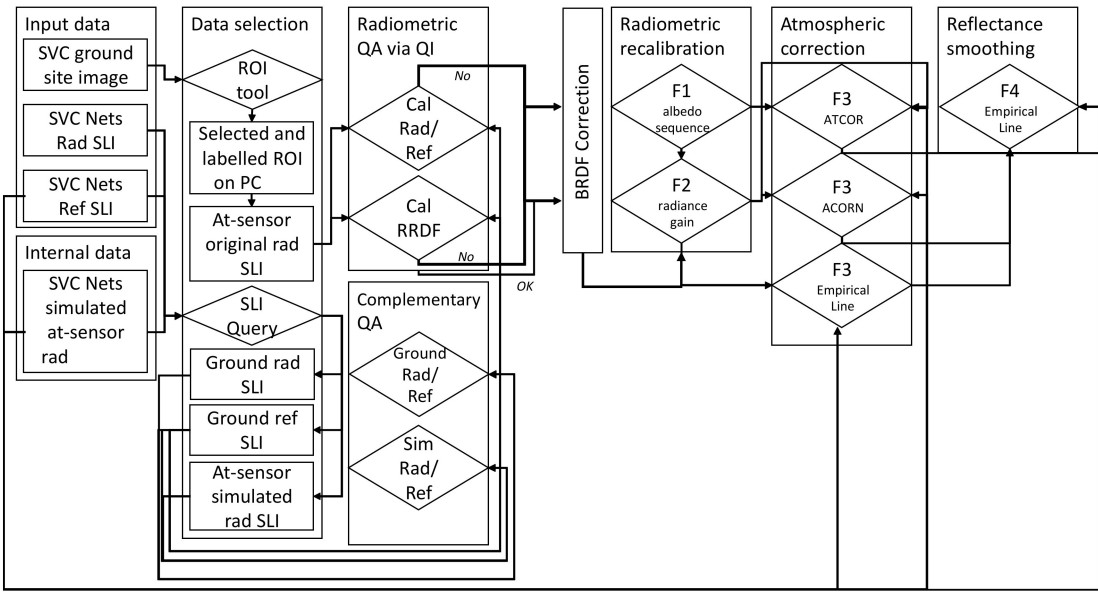

**Figure 1.** SVC scheme.

According to the suggested approach, the solution is calculated in the following steps:

(1) Radiometric quality indicators—the first step is dependent on a selected region of interest by the operator. This step is performed on the UAS-based orthophoto and projected on the reconstructed pout cloud.

Prior to extracting the above-mentioned coefficients, it is important to inspect the sensor's radiance performance using quality indicators (QI). The most common radiometric investigation uses MODTRAN to reconstruct the atmosphere above the sensed surface and then compare the results with the obtained at-sensor radiance (Secker et al., 2001).

The UAS multispectral data are collected under operational conditions and thus the simulated AOT might affect the SVC results. Therefore, an additional preparation stage is needed [61]. The procedure requires atmospheric model applied in an iterative mode that will be completed only by achieving reliable results. Alternatively, the at-sensor radiance can be examined by its corresponding reflectance without applying any radiative model [60] using the radiance to reflectance ratio (Rad/Ref) and radiance-to-reflectance difference ratio (RRDF) QIs.

According to the calibration theory, the at-sensor measured radiance equal to the sun's radiance at nadir zenith angle times the atmospheric transmittance coefficient adding the selective scattering contribution. The first QI is termed Rad/Ref and calculated by dividing the at-sensor radiance by the surface reflectance coefficient of a selected ground target (e.g., SVC nets). The second QI is assessing the quality of the radiometric output using at least one set of net measurements and calculating the RRDF. The RRDF ratio is invariant with respect to the surface reflectance, thus, all calculated responses must be identical.

(2) BRDF correction—following the recommendations reported by [64], prior to submitting the imagery data to the radiometric recalibration (F1 stage), the BRDF effect must be estimated and reduced. This essential stage was included in the modified scheme of the SVC method to provide more realistic at-sensor radiance data. Once the point/pixel/surface is facing the sensor (in nadir), the calculated angle is equal to 90°.

The angle decries when the point/pixel/surface is tilted then its radiance is scattered and reflected in an off-nadir way. The calculated ground target depression angle is used to correct solar information (azimuth and zenith), which is calculated by a given date, time, and geographic location at a central given coordinate. The corresponding solar information is used to retrieve the BRDF correction coefficients (Rcorr) for the SVC calibration nets target [64]. The calculated coefficients are further applied for the full scanned scene, the full point cloud data.

(3) The SVC correction-the at-sensor radiance is converted into accurate reflectance by applying four stages: normalization of the albedo sequence (F1) inspected by QIs, radiometric calibration gain using the net ground-truth reflectance (F2), applying a model-based atmospheric correction (F3) using ATCOR5 model, ACORN and empirical line method, and spectral polishing using the net ground0truth reflectance (F4). The SVC scheme is guided by the QIs scores. Well-calibrated sensors can proceed directly to stages F3 and F4. When the Rad/Ref holds a theoretical sequence but the RRDF indicator gives an indistinct result, the F2 stage should be applied before stages F3 and F4. Finally, when both Rad/Ref and RRDF indicators generate indistinctly, the full SVC correction chain is necessary, i.e., F1 and F2 until both parameters (Rad/Ref and RRDF).

### 2.4. Data Processing and Analysis

### 2.4.1. Spectral Model for TC Content

C and N content in the top-ash layer was quantified. While organic C is measured using elemental analyzers, many studies, e.g., [59,65,66] reported that DR-FTIR spectroscopy yields good estimates for PyC. To quantify the area of the aromatic and aliphatic n(CH) bands, the spectra were baseline corrected in the 3200 to 2700 $cm^{-1}$ region using a quartic fitting function. The integrated area of the aromatic n(CH) bands was determined by spectral integration of the FTIR spectra between 3150 and 3000 $cm^{-1}$, and the aliphatic n(CH) bands were determined based on the area under the curve in the 3000 to 2750 $cm^{-1}$ region.

The partial least squares regressions (PLSR) based model developed by [67] for fuel type, heating/combustion temperature and respective C content of residues was applied. This approach can accurately predict TC content (in %) of 'real world' samples based on DR-FTIR spectra. In practice, the area under each of the individual peaks relative to aromatic rings in amorphous carbon at 1590–1600 $cm^{-1}$, carbonate at 1440–1450 $cm^{-1}$, and a carboxyl group at 1700–1720 $cm^{-1}$ was calculated. All measured spectra characterized by a broad signal in the region 4000–2500 $cm^{-1}$, attributable to strong stretching vibration of OH groups in Al-OH (3600–3500 $cm^{-1}$) and hydroxyl at around 3400 $cm^{-1}$. A shoulder between 3082 and 3066 $cm^{-1}$ was assigned to C-H stretching in alkanes and/or aromatic rings (=CH). The asymmetric (2930 $cm^{-1}$) and symmetric (2840 $cm^{-1}$) CH stretching appeared in all spectra, therefore the peaks in the range 1240–1200 $cm^{-1}$ associated to C-N and phenolic C-O stretching were selected for further analysis. The region 2400–2000 $cm^{-1}$ showed peaks to assign to the CN stretching vibration of nitrile and cyanimide groups [68].

### 2.4.2. Partial Least Squares Discriminant Analysis and Machine Learning for TC Content

The input training data (sample from the field, and all laboratory samples) is presented as an unfolded matrix where the rows represent observations (spectral measurements in three different datasets (varies in spectral resolution): OceanOptics USB4000 portable spectrometer (across VNIR spectral region), RedEdge-MX Micasense multispectral UAS-based imagery and severity map calculated by Char Index based on color coordinates (Brightness Index of DJI Phantom 4 Pro RGB images) and columns represent the true classes (TC content of residues calculated based on DR-FTIR spectra via PLSR model). Since spectral image enables quantification and classification at the individual pixel level, the following classification approaches were applied and then evaluated. The proposed methods were implemented in a MATLAB computing environment (release R2019b, The MathWorks, Inc., Natick, MA, USA).

PLS-DA classifier, a well-known classification method for spectral imaging datasets, was used to build spectral models. Since model performance is highly connected to the selection of the appropriate number of latent variables, cross-validation was applied to determine the optimal number by ranking the evaluation of the correct classification rate. Normally the accuracy increases rapidly for the first few latent variables and then remains relatively constant. However, a wrong number of latent variables is unsatisfactory as it will result in under or over-fitting of the data and poor model performance.

Artificial Neural Networks (ANN) is a system modeled on the human brain. It consists of a large number of interconnected processing nodes called neurons, structured in different layers of varying numbers, enabling the system to process multiple inputs from external sources. The heuristic approach was adopted for multi-layer perceptron. It has been used for back-propagation training of feed-forward neural networks and utilized in several real-life applications such as prediction and estimation. Feed-forward back-propagation networks were developed with the training functions Levenberg–Marquardt and Bayesian regularization. A grid search with two tuning parameters (the number of nodes in the hidden layer from 3 to 20, and the decay of weight at each iteration set at 0.01, 0.05, and 0.1) was used to select the model with the lowest RMSEP values.

The input of the 1-D CNN is a spectrum one-dimensional vector, therefore the first step was to extract the spectral vector from the multispectral image by unfolding the images (rows by columns over bands). The function of the convolutional layer is to convolve the input data by applying sliding convolutional filters and producing the convolved features as the output also known as a feature map. Each type of extracted feature was generated by a convolutional kernel. Usually, the kernel is moved from left to right and then from top to bottom over the input with a step of 1. Stride convolution has a larger user-defined step size for traversing the input. In 1-D CNN, the convolution kernel and feature map are both one-dimensional. The convolution extracts feature according to Equation (2).

$$x_k^i = \sum w_k^{i,c} * x^{i-1,c} + b_k^i \tag{2}$$

where the $i$th layer, $k$ is the index for a specific feature map, $c$ refers to the band number of the input $x^{i-1}$, $w_k{\char`\^}(i,c)$ is the kth convolution kernel corresponding to the $c$th band, $b_k{\char`\^}i$ refers to the bias of the $k$th feature map.

Batch normalization (BN) was carried out before activation to avoid distribution shift by applying transformation that maintains the mean of the convolved features close to zero and the variance of the convolved features close to one [69]. It normalizes its inputs $x_k{\char`\^}(i-1)$ the input at kth feature map via the computed mean μ and variance $\sigma^2$ of a mini-batch and over each input band (Equation (3)).

$$\hat{x_k^i} = \frac{x_k^i - \mu}{\sqrt{\sigma^2 + \varepsilon}} \tag{3}$$

where $\varepsilon$ improves numerical stability in small mini-batch variance where inputs with a mean of zero and variance of one are not suitable for the subsequent layer.

In this case, the BN layer can be shifted and scaled. The offset and scale factors are learnable parameters that are updated during network training. The normalized features are input into a layer with rectified linear unit (ReLU) activation function (F(x) = max(0, x), and a dropout layer is then applied to prevent overfitting the model. The choices of dropout neurons are random with a given probability, defined by the user. After the dropout layer, a fully connected (FC) layer is used to merge all feature maps. Therefore, the number of neural nodes depends on the convolution kernel size, the sampling kernel size, and the number of feature maps.

Since a multi-classification task was performed a softmax layer was performed after the last FC layer. The input of softmax comes from k different neurons of the FC layer.

In order to select proper parameters for the model, the influence of filter size, number of filters, and stride on performance was assessed. The filter size was gradually increased

from 5 to 50 at a step of 5, and the 1-D CNN models were developed keeping the other parameters constant (e.g., number of filters and fixed stride). The model with the best classification result at the convolution kernel size was selected (by plotting classification accuracy against filter size for the validation set). Finally, the optimal stride was defined using the conditions already determined by filter size and the number of filters. Generally, the size of the stride needs to be smaller than the filter size, therefore, the models are built with an increase of between 1 and 5 steps.

The proposed method was compared to a conventional and popular remote sensing-based index that highlights and maps burned regions based on spectral imagery. The burn area index (*BAI*) identifies burned land in the red to near-infrared regions of the electromagnetic spectrum on atmospherically free data, by emphasizing the charcoal signal in post-fire images. The index is computed (Equation (4)) from the spectral distance from each pixel to a spectral reference point, where recently burned areas converge. It has been identified as one of the best indices to map burnt areas [70] and maps four fire severity classes: unburnt, low-, moderate- and high fire severity.

$$BAI = \frac{1}{(0.1 - Ref_{650nm})^2 + (0.06 - Ref_{800nm})^2} \tag{4}$$

A new UAS-based index called the Char Index was proposed by Smith (Smith et al., 2005), to highlight charred organic surfaces (Equation (5)). This is a composite index based on observation and quantified using a *Brightness* Index and a flat reflectance spectrum across the VIS region (Maximum *RGB* Difference Index *MaxDiff* or lack of colour). The results of the Char Index (CI) were categorized into severity by predefined thresholds [55].

$$\text{Char Index } = Brightness + (MAxDiff * 15) \tag{5}$$

*2.5. Validation and Verification*

The spectral information measured by portable spectrometers (across the VNIR spectral region) in situ and under laboratory conditions assisted in the calibration and validation of the UAS-based spectral imagery. To compare how well the spectra in the multispectral UAS-based imagery match the validation spectra collected in situ, the spectral angle mapper (SAM) metric [71] was calculated. SAM compares spectral signatures by calculating the angle between two spectral vectors in the n-dimensional feature space. Three validation spectra and the corresponding image spectra are collected and, subsequently, averaged for each validation site within the study area. The spectral comparison was performed between two mean spectra at each validation location. The effect of spectral and spatial resampling on spectral similarity is further presented and discussed.

The comparison between TC measured via Leco TruSpec CHN and DR-FTIR spectra were performed using Student's *t*-test to measure the variability for group A: the analytical TC content (CHN) and group B: the spectral based TC content (DR-FTIR). The null hypothesis was that there was no difference between the groups.

The mean squared error (*MSE* in Equation (6)) and mean absolute error (*MAE* in Equation (7)) between the expected values and the true values was estimated and reported.

$$MSE = \frac{1}{n} \sum_{t=1}^{n} (x_t - \hat{x}_t)^2 \tag{6}$$

$$MAE = \frac{\sum_{t=1}^{n} |x_t - \hat{x}_t|}{n} \tag{7}$$

where, $\hat{x}_t$ is the predicted TC content (in %) of the model's *t*th observation (step), $x_t$ is the targeted one, and $n$ indicates the number of samples.

Classification models were built on a training set and then applied to a validation set (comprising spectra extracted from each dataset before training), to enable comparison of model performance. In addition, a test set, using an experimental field site, was used for

model evaluation. Primarily, the performance of the developed model was assessed by the classification accuracy, also known as correct classification rate (CCR).

## 3. Results

The spectra comparison was calculated using the SAM approach. SAM for image and ground validation spectra across the VNIR region were collected using USB400 OceanOptics at each sampling spot. The ground truth VNIR spectra were resampled into 5 bands of the RedEdge-MX Micasense camera and 3 bands (RGB) of the DJI Phantom4, according to its spectral configuration (central wavelength and bandwidth).

Student *t*-test results show insignificant variation in the analytical and spectral-based TC content of 24 selected samples (F 0.71, *p*-value 0.41), confirming that there is no difference between the analytical and the spectral-based protocols used to determine TC.

The effect of data calibration steps is presented in Figure 2. All examined datasets responded to the radiometric calibration in general and to the BRDF correction in particular, with increasingly higher similarity (lowering of spectral angle values) from raw (DN) data with the greatest dissimilarity to data corrected via SVC with BRDF correction. Moreover, the level of heterogeneity in the data gradually decreased after applying the SVM method and then implementing the BRDF correction, as shown by the reported error bars (minimum and maximum values) getting closer to the average spectral angle. These impacts were more pronounced for the multispectral dataset collected by the RedEdge-MX Micasense camera than the RGB imagery data collected by the DJI Phantom4, this is due to its limitation in the spectral domain (only three wide bands across the VIS region). Likewise, the level of heterogeneity in all examined datasets reported as raw data (Figure 2) was greater in the data collected by the DJI Phantom4.

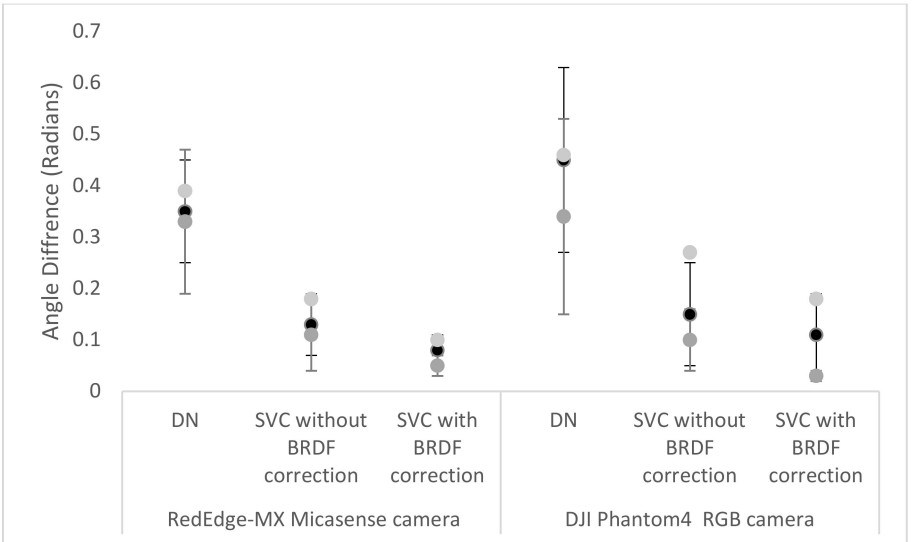

**Figure 2.** The similarity of image spectra (as raw data, partially calibrated data based on SVC, and fully calibrated data SVC with BRDF correction) and ground validation spectra derived from SAM for the three examined datasets: the laboratory dataset in light grey, the control field experiment in grey, and the urban wildfire site in black. The error bars show minimum and maximum values for each dataset.

The classification model performance of PLS-DA calculated training datasets and the validation and test datasets are shown in Tables 2 and 3, respectively.

**Table 2.** Performance measurements of the PLS-DA model.

|  |  | LVs | RMSE | MAE | $R^2$ |
|---|---|---|---|---|---|
| Spectrometer | Reflectance | 15 | 0.08 | 0.07 | 0.989 |
| RedEdge-MX Micasense camera | DN | 5 | 1.32 | 1.11 | 0.594 |
|  | SVC without BRDF correction | 4 | 1.18 | 0.92 | 0.874 |
|  | SVC with BRDF correction | 4 | 0.41 | 0.32 | 0.923 |
| DJI Phantom4 RGB camera | DN | 3 | 1.50 | 1.43 | 0.588 |
|  | SVC without BRDF correction | 3 | 1.21 | 1.13 | 0.752 |
|  | SVC with BRDF correction | 3 | 0.98 | 0.84 | 0.855 |

**Table 3.** PLS-DA model performance for validation and prediction of TC content reported as correct classification rate (CCR in %).

|  |  | Validation | | | Test | | |
|---|---|---|---|---|---|---|---|
|  | Dataset | Laboratory | Controlled Field Experiment | Urban Wildfire | Laboratory | Controlled Field Experiment | Urban Wildfire |
| Spectrometer | Reflectance | 98.12 | 99.84 | 96.39 | 99.14 | 96.72 | 94.92 |
| RedEdge-MX Micasense camera | DN | 59.42 | 59.86 | 59.73 | 57.81 | 57.29 | 58.11 |
|  | SVC without BRDF | 93.67 | 87.67 | 64.55 | 92.18 | 81.44 | 63.91 |
|  | SVC with BRDF | 95.94 | 92.45 | 91.16 | 91.78 | 90.84 | 89.57 |
| DJI Phantom4 RGB camera | DN | 58.77 | 49.06 | 47.82 | 49.94 | 49.27 | 40.81 |
|  | SVC without BRDF | 80.73 | 78.59 | 69.61 | 79.68 | 73.64 | 66.21 |
|  | SVC with BRDF | 87.52 | 83.61 | 70.63 | 88.42 | 85.31 | 72.09 |

According to Table 2, the full reflectance (601 wavelengths) model with 15 LVs showed the best performance with values of 0.08, 0.07, and 0.989 for RMSE, MAE, and an $R^2$, individually. The multispectral model (5 bands) achieved a relatively low RMSE of 0.41, MAE of 0.32, and a relatively high $R^2$ (0.923) for fully corrected and calibrated data (SVC with BRDF correction) with only 4 LVs. The RGB imagery data showed the weakest performance with an $R^2$ of 0.855 and a relatively high RMSE of 0.98, and an MAE of 0.84.

Overall, as reported in Table 3, the prediction results for the test set (i.e., TC content in %) were slightly lower than that of the validation set. It is important to note that the pre-processed and radiometrically corrected images improved the accuracy of the test set, for example from CCR of 59.73% for raw data to 91.16% for SVC and BRDF corrected multispectral data (RedEdge-MX Micasense camera) as well as 47.82% for raw data and 80.63% for severity map based on CI calculated by Brightness index (based on raw and calibrated RGB DJI Phantom4 imagery) at the urban wildfire site.

The average of the summed RMSEP was recorded for each neuron, and the model with the lowest RMSEP value (Figure 3) was considered the best. This process was repeated using three inputs for validation and test datasets simultaneously: laboratory, controlled field experiment, and urban wildfire site, for each sensor: OceanOptics USB4000 portable spectrometer, RedEdge-MX Micasense multispectral UAS-based imagery, and DJI Phantom 4 Pro RGB images, in order to compare the performance of the spectral image datasets. The aforementioned steps in ANN were carried out using the Levenberg–Marquardt and Bayesian regularization training functions to find out which one was better suited for this purpose. The results for reflectance spectra (measured by spectrometer), multispectral, and

CI data show that networks with 3 to 5 neurons performed poorly, and the addition of a 6th neuron greatly improved the models. Adding more neurons was beneficial until the network had 9–10. The error of the models slightly increased at the 10 neuron threshold, implying that 6 neurons were sufficient for TC content prediction, with more neurons producing overfitted models. The performance of the Levenberg–Marquardt-based ANNs was inferior to the Bayesian regularization models for all examined criteria. Despite the longer time needed to train the Bayesian regularization models, the use of the Levenberg–Marquardt model remains unjustified (Figure 3).

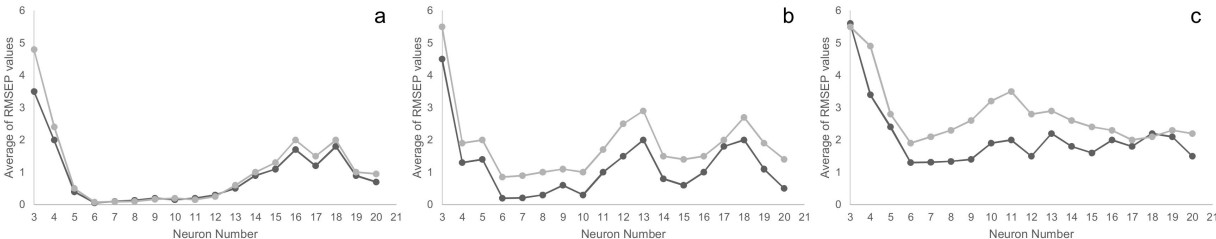

**Figure 3.** Average RMSEP prediction values using TC content estimations from (**a**) reflectance spectra (portable spectrometer), (**b**) multispectral (raw and calibrated RedEdge-MX Micasense imagery) and (**c**) CI severity data (calculated using raw and calibrated RGB DJI Phantom4 imagery) input to ANN with Levenberg–Marquardt (bright grey plot) and Bayesian regularization (dark grey plot) models.

The validation performance (regression plots with R-scores) for the RedEdge-MX Micasense camera with and without the BRDF correction and for the DJI Phantom4 RGB camera with and without the BRDF correction, on 15% of the input data (95 spectra) are reported in Figure 4.

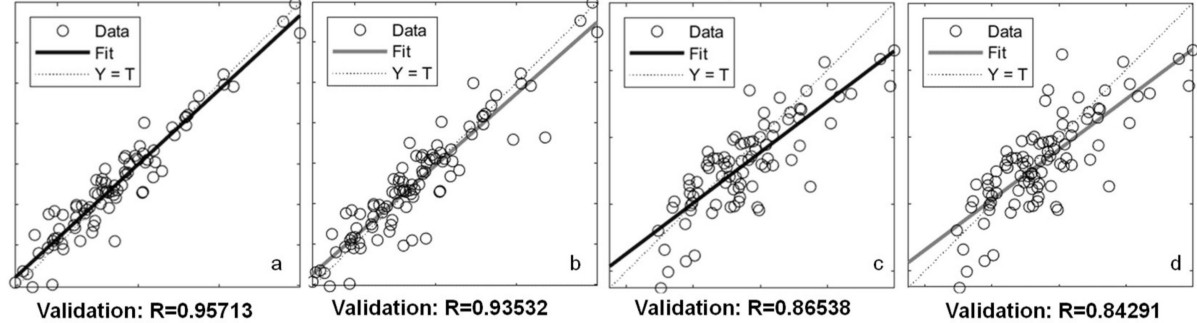

**Figure 4.** Regression plots for the ANN validation with the Bayesian regularization on Laboratory data where (**a**) is the RedEdge-MX Micasense camera with the BRDF correction, (**b**) is the RedEdge-MX Micasense camera without the BRDF correction, (**c**) is the DJI Phantom4 RGB camera with the BRDF correction and (**d**) is the DJI Phantom4 RGB camera without the BRDF correction.

The TC predicted by ANN with the Bayesian regularization model was compared to the measured TC by calculating the CCR (Table 4).

The ANN model was validated by comparing its predictions to those of a PLS-DA model built using the same data and inputs. The main difference between the performance of the ANN and PLS-DA models was the ability of ANN to recognize the nonlinear effect of TC content and spectral data. Of note is that the severity data calculated by CI (calculated on raw and calibrated RGB DJI Phantom4 imagery) was too coarse. Since the multispectral imagery data collected by RedEdge-MX Micasense camera and processed via SVC with BRDF correction gave the most accurate results, these data were further examined and subjected to a 1-D CNN model.

**Table 4.** ANN model performance for prediction of TC content reported by RSMEP and correct classification rate (CCR in %).

| Dataset | | Lowest RMSEP for Bayesian Regularization | Lowest RMSEP for Levenberg–Marquardt | Test | | |
|---|---|---|---|---|---|---|
| | | | | Laboratory | Controlled Field Experiment | Urban Wildfire |
| Spectrometer RedEdge-MX Micasense camera | Reflectance | 0.06 | 0.08 | 98.28 | 98.3 | 96.2 |
| | DN | 1.2 | 1.35 | 64.31 | 61.38 | 53.65 |
| | SVC without BRDF correction | 0.98 | 1.2 | 88.29 | 87.67 | 79.62 |
| | SVC with BRDF correction | 0.26 | 0.96 | 92.11 | 90.52 | 91.48 |
| DJI Phantom4 RGB camera | DN | 1.5 | 2.4 | 66.72 | 41.27 | 43.84 |
| | SVC without BRDF correction | 1.8 | 2.1 | 82.91 | 81.49 | 81.43 |
| | SVC with BRDF correction | 1.3 | 1.9 | 84.28 | 83.17 | 82.67 |

Tuning parameters for DL requires extensive processing. In order to select suitable parameters for the 1-D CNN model, the effects of the filter size, number, and stride on performance were studied. Filter size was gradually increased from 5 to 50 with an interval of 5, and the 1-D CNN models were developed while keeping the other parameters constant (e.g., number of filters = 20, stride = 1). The classification accuracy for the test samples plotted against filter size is seen in Figure 5a. The best classification was found when the filter size was set to 15 and this was used for the subsequent tuning of the other parameters. The number of filters ranged from 5 to 30 increasing by 5 while keeping the other parameters constant (e.g., filter size = 15, stride = 1). The accuracy of the test samples was plotted against the number of feature maps in Figure 5b. The accuracy increased rapidly until 15 filters and then decreased. The optimal stride (sampling step size), was determined for 15 filters. The models were built by increasing the stride in increments of 1 from 1 to 5. As shown in Figure 5c, the best classification performance was achieved when the stride was set to 2.

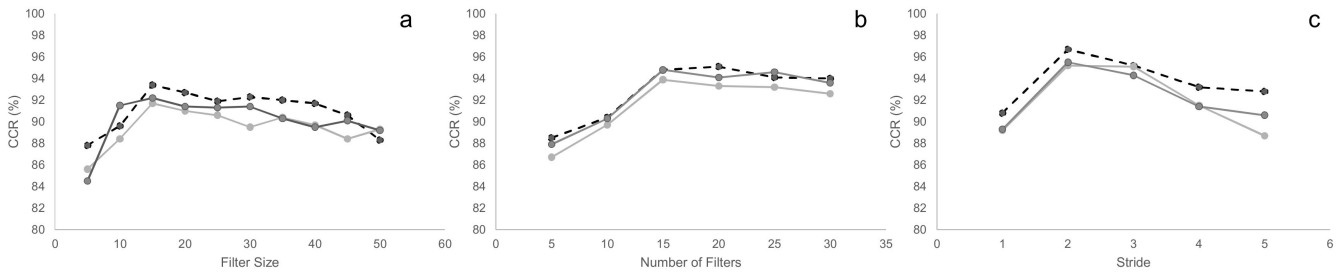

**Figure 5.** Classification performance of validation set (all sites at once) with different filters: (**a**) sizes; (**b**) numbers; (**c**) strides for the following datasets: reflectance spectra (black dashed line), multispectral calibrated RedEdge-MX Micasense data (dark grey line) and severity data based on CI calculated on calibrated RGB DJI Phantom4 data (light grey).

As seen in Figure 6a,b the loss on the training set decreased rapidly for the first 25–30 iterations, suggesting that the network was learning to classify TC spectra quickly. The loss of the validation set decreased as fast as the training loss, implying that this model generalizes reasonably well to unseen data. We also noted that high accuracy (above 90%) for training data was achieved with much less loss after approximately 30 iterations.

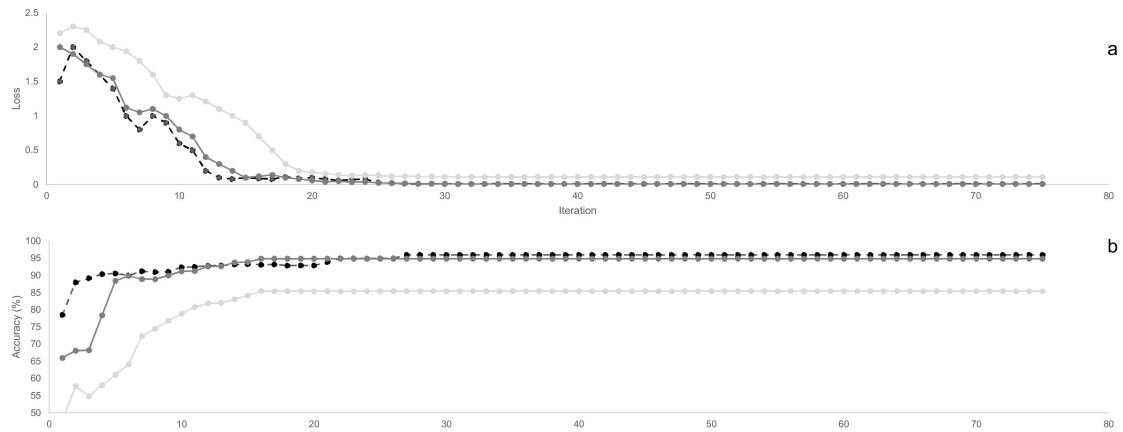

**Figure 6.** Training progress of 1-D CNN with (**a**) loss and (**b**) accuracy plotted against iteration (number of epochs) for the following datasets: reflectance spectra (black dashed line), multispectral calibrated RedEdge-MX Micasense data (dark grey line) and severity data based on CI calculated on calibrated RGB DJI Phantom4 data (light grey).

The classification results were compared with the previous approaches described. The 1-D CNN showed higher accuracy with a CCR of 96% for reflectance spectra data, 95.4% for multispectral calibrated RedEdge-MX Micasense data (dark grey line), and 85% for severity data based on Char Index (CI) calculated on calibrated RGB DJI Phantom4 data from test samples from the urban wildfire site.

Figure 7 shows test samples from the controlled field experiment compared to the results of BI (Figure 7a), PLS-DA model (Figure 7b), ANN model (Figure 7c), and 1-D CNN model (Figure 7d). The BI map (Figure 7a) is a binary map of post-fire pixels that can only be compared to the ground truth and other three products by calculating intersecting areas, without considering fire severity. The results for the PLS-DA model, ANN model, and 1-D CNN applied to the same data (Figure 7b–d) show that 1-D CNN has better classification performance (higher CCR and lower FDR in Figure 7).

The results of the experimental (control) dataset in Figure 8 show the rates of success for PLS-DA, ANN, and 1-D CNN. The PLS_DA had the lowest performance rate, and 1-D CNN applied to the same data improved classification performance. The greatest confusion in 1-D CNN was within the second class (10% TC) with a success of 91.4%.

The 1-D CNN results were compared with the severity map calculated by the CI (Figure 9). The area of the charred surface was similar indicating the extent of burning in the surface organic layer. The CI corresponded to the presence of light-colored ash, which indicates that the fire completely combusted the organic material. However, the ash, unlike char, was misclassified and mixed with rocks, as the index was only based on color. The high degree of flattening/coarseness in severity data makes it relatively less useful than UAS data, compared to the developed TC model. This is mainly due to the fact that the remote sensing indices were primarily developed for coarser spatial resolution, therefore have a negligible response to higher resolution data. According to Figures 9 and 10, it is clear that both BAI and CI indices are not informative and too rough (e.g., no details, generally patchy representation) for UAS-based multispectral data, which can quantitatively map biochemical properties using appropriate spectral information and high-resolution imagery data, as shown in Figure 9.

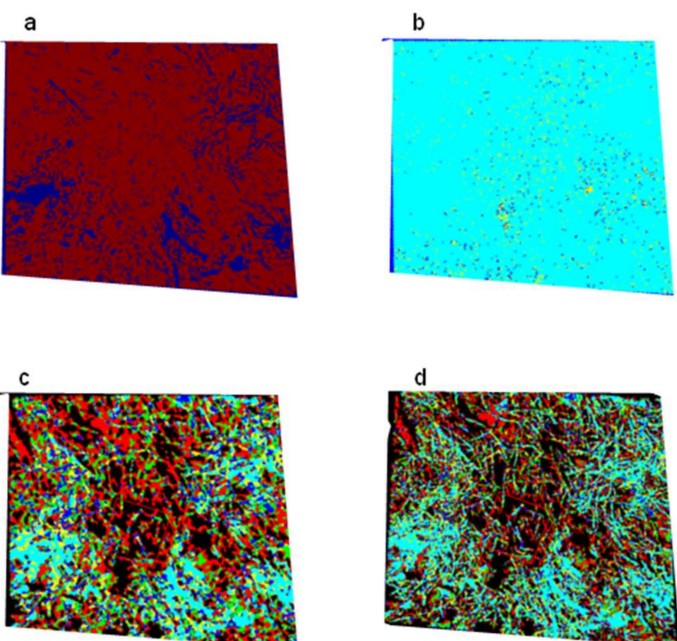

**Figure 7.** TC classification performance of models on the experimental (control) dataset (8 of 22 two sites of 2 m², were test sites and one selected subplot was plotted) by (**a**) BI, (**b**) PLS-DA, (**c**) ANN (**d**) 1-D CNN, according to five classes: 20–17% (cyan), 17–16% (yellow), 16–15% (blue), 15–14% (green), 14–10% (red) on multispectral calibrated RedEdge-MX Micasense image.

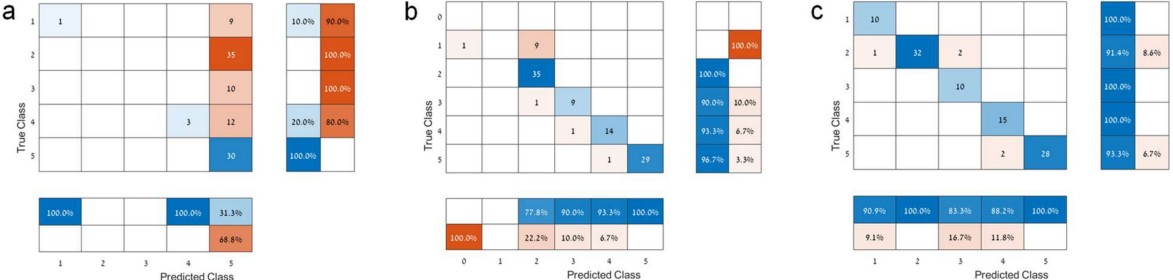

**Figure 8.** Confusion matrix for TC classification maps produced by (**a**) PLS-DA, (**b**) ANN, (**c**) 1-D CNN models on the control dataset (eight subplots) using multispectral calibrated RedEdge-MX Micasense images.

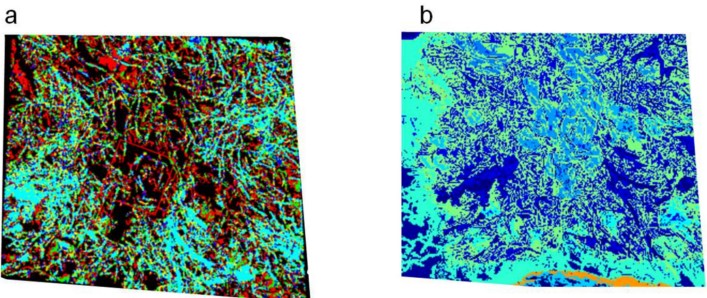

**Figure 9.** 1-D CNN based TC (**a**) five class map: 20–17% (cyan), 17–16% (yellow), 16–15% (blue), 15–14% (green), 14–10% (red), versus the CI (**b**) severity categories: very high (blue), high (yellow), moderate (cyan) and low (orange), performed on a multispectral calibrated RedEdge-MX Micasense image.

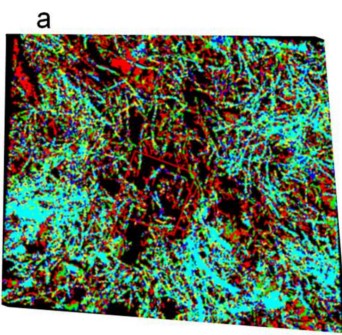
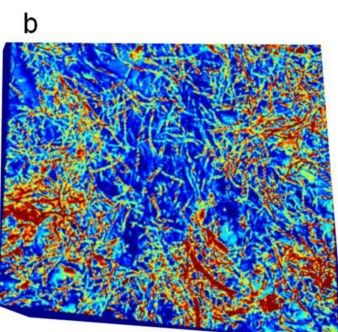

**Figure 10.** 1-D CNN based TC (**a**) five class map: 20–17% (cyan), 17–16% (yellow), 16–15% (blue), 15–14% (green), 14–10% (red), versus (**b**) continuous stretched quantitative mapping of TC content (blue to red–10% to 20%).

The results of urban wildfire ROIs in Figure 11 show similar rates of success, PLS-DA had the lowest performance rate, and 1-D CNN applied to the same data improved classification performance. The greatest confusion was between the second and the third classes in both PLS-DA and ANN models (10% to 15% TC).

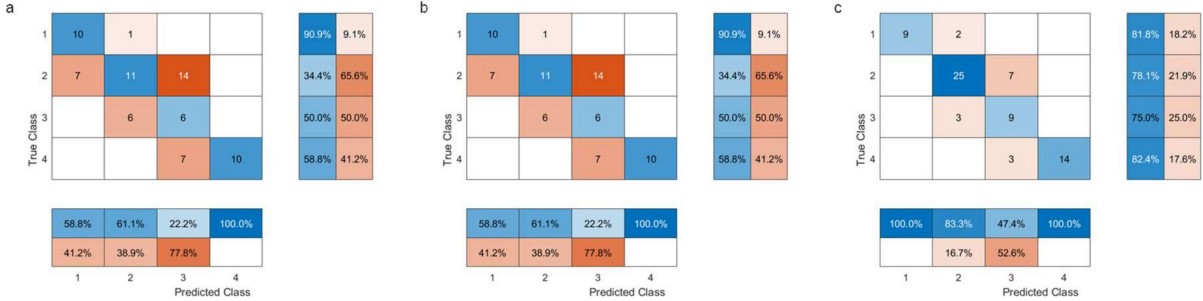

**Figure 11.** Confusion matrix for TC classification maps produced by (**a**) PLS-DA, (**b**) ANN, (**c**) 1-D CNN models on the urban wildfire test dataset (11 subplots) using multispectral calibrated RedEdge-MX Micasense images from four classes: (1) <10%, (2) 10–14%, (3) 14–15%, (4) >15%.

The 1-D CNN continuous stretched quantitative mapping of TC content shows a strong linear and even stronger polynomial correlation in four fire severity categories (very low to high), but not for the very high fire severity category (Figure 12). A positive linear and second-degree polynomial relationship between the suggested method and the CI fire severity and the fact that high severity fire was not observed in the field verified the spectral model accuracy as a metric of actual fire severity.

The burn severity map based on CI calculated using brightness data was compared with the proposed TC content model using the full urban wildfire scene (Figure 13). The coarseness of severity categories and their general patchiness versus highly detailed and more accurate (according to the confusion matrix in Figure 10) TC content map is illustrated. However, it is important to note that both methods could recognize and extract the burned area equally well, without getting confused by the living biomass, in particular trees and shadows.

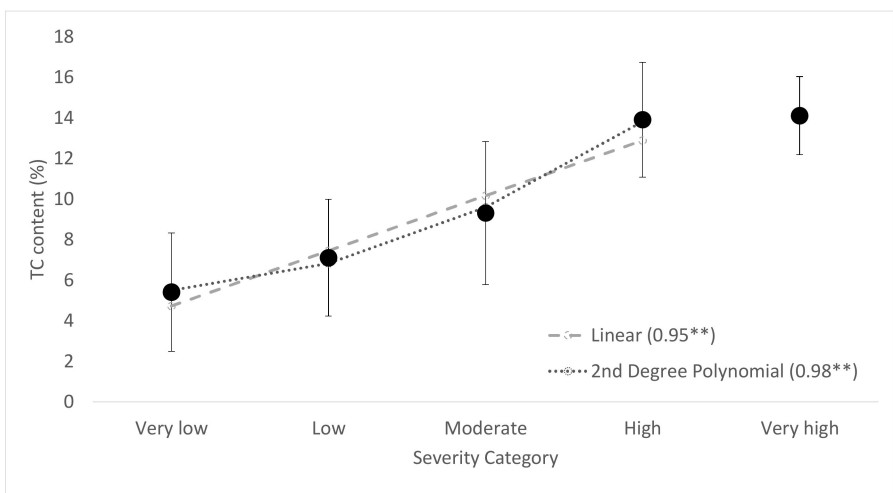

**Figure 12.** Average TC content (with standard deviation STD) for each severity category based on the intersection between the CI severity categories and 1-D CNN continuous stretched quantitative mapping of TC content. The dashed grey line shows the linear correlation ($R^2$ 0.95 with *p*-value 0.0004) and the dotted black line shows a second-degree polynomial function ($R^2$ 0.98 and *p*-value 0.000) between TC content and Severity Category up to High-Severity.

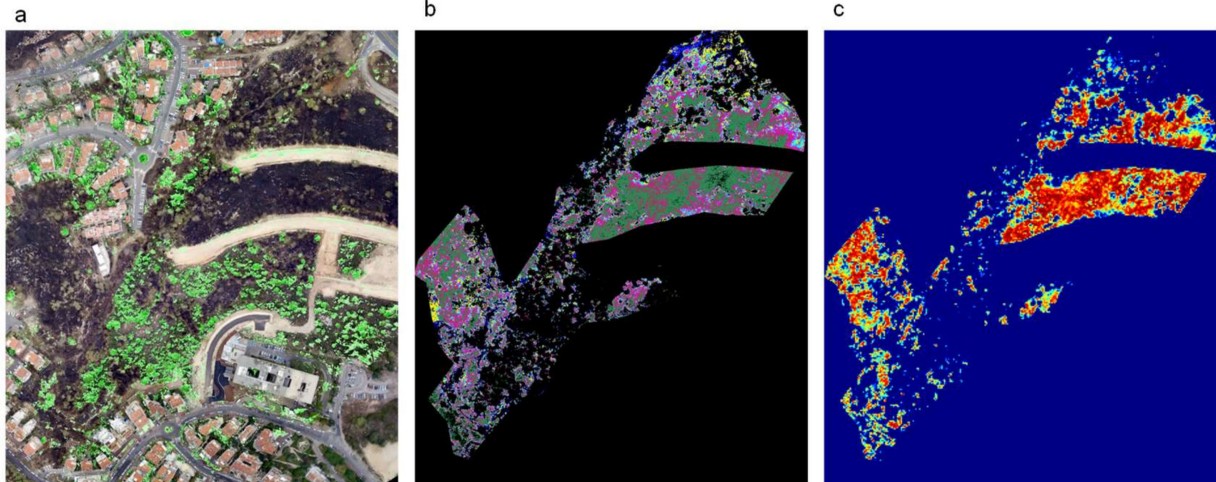

**Figure 13.** UAS-based orthophoto with vegetation layer (in green) calculated by (**a**) NDVI index, (**b**) CI severity categories: very high (maroon), high (sea green), moderate (cyan), low (yellow), very low (blue), (**c**) 1-D CNN continuous stretched quantitative mapping of TC content (blue to red—5% to 18%).

## 4. Discussion and Conclusions

Efforts to define and classify fire severity only began recently. All of the currently used metrics have strengths and weaknesses when evaluating the type and magnitude of fire effects within a specific ecosystem.

The main disadvantage of the most widely held remote sensing approach, known as the spectral indices, is its moderate accuracy for mapping the post-fire environment in general, and fire severity of a single fire in particular (60–70% accurate compared to field validation [38,39]). Furthermore, since, fire severity indices differentiate between photosynthetic and non-photosynthetic ground targets they are considering many pre-fire conditions (e.g., vegetation structure, moisture, soil type, and topography [43,44]). The main advantage is that those indices are generally accepted as robust methods for determining fire scars in the landscape and assessing the recovery of the vegetation [40–42] with a limited ability to demonstrate the spatial variability and patchiness. However, all

the efforts categorizing the values of spectral indices into standardized severity groups are mostly coming to failure.

Our study demonstrates that TC can be mapped with high accuracy (mean accuracy of 93% across all classes) using a spectral model derived from multispectral calibrated RedEdge-MX Micasense images with 1-D CNN. In addition, classification accuracy was very high (>95%) for unburnt and higher severity classes but slightly lower for the lower severity classes (83–88%). These findings agree with previous studies using satellite imagery (Landsat and Sentinel 2), showing that machine learning classifiers are well suited to the broad-scale mapping of fire severity with remotely sensed imagery [48,72,73]. However, even the most recent studies are not sensitive to local scale and spatial patchiness [74,75].

The practical application of the severity maps in fuel assessment and fire behavior analysis requires a clear understanding of how mapping approaches perform across different severity classes. Using spectral information and 1-D CNN classification of UAS-based multispectral imagery reflectance, our approach produced a highly accurate map of unburnt and high severity wildfires in landscapes with high to moderate canopy density and variable topographic roughness.

The study in [7] emphasized the limited ability of remote sensing to predict ecosystem response and recommended using field studies to improve the interpretation of ecosystem responses from severity levels acquired using remote sensing. Our study shows that estimating post-fire severity with color or CI does not give a complete and accurate picture as areas classified as very high severity by ash color were not necessarily associated with extremely high burning temperatures and in fact, were wrongly categorized.

The higher spatial resolution of UAV multispectral data allows detailed mapping with almost no unmixing effect. The high spatial resolution improves the classification of low severity fire in areas with a dense canopy. Advanced ML in 1-D CNN helped overcome the limitations of the pixel-based approach used in this study, particularly for improving the classification of low and moderate severity fires.

The proposed TC content model applied to UAV multispectral imagery offers an alternative to satellite-based approaches (calculated via spectral/vegetation indices) to map fire severity.

**Author Contributions:** A.B., L.W., D.R., C.I. and N.S.-Z. were responsible for conceptualization, supervision, and fund acquisition; S.H. was responsible for data collection; A.B. was responsible for methodology development, software development, code, validation, writing, and editing; L.W. was responsible for implementation and manuscript review. All authors have read and agreed to the published version of the manuscript.

**Funding:** This research was supported by Grant 2014299 from the United States-Israel Binational Science Foundation (BSF).

**Acknowledgments:** We gratefully acknowledge the European Cooperation in Science and Technology (COST) action CA18135 "FIRElinks" (Fire in the Earth System: Science & Society), and CA16219 "HARMONIOUS" (Harmonization of UAS techniques for agricultural and natural ecosystems monitoring) for technical support.

**Conflicts of Interest:** The authors declare no conflict of interest. The funders had no role in the design of the study; in the collection, analyses, or interpretation of data; in the writing of the manuscript; or in the decision to publish the results.

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
