# Peer review of "Total Carbon Content Assessed by UAS Near-Infrared Imagery as a New Fire Severity Metric"

_remotesensing, doi:10.3390/rs14153632_

Round 1

Reviewer 1 Report

Review report for the paper “Total Carbon content assessed by UAS near-infrared imagery as a new fire severity metric”; remotesensing-1787878.

Significance :

. The scientific content of this paper is correct. 
. The technical quality of this paper is correct. 
. The results could be better presented. This would emphasize the quality of the presented work.

. The limits of the results obtained in this paper are not mentioned. This point should be investigated. 
Quality of presentation:

. The abstract is clear and presents correctly the subject addressed in this paper. 
. Introduction - What are your contributions?

. Why did you use ML - NN? Why not other tools in the ML filed?

. How we can judge about quality of the training set?

. Better highlight novelty in the study.

. Better define motivations for the research.

. The data and analyses should be better presented. Add more discussion on the results. Add comparisons with existing approaches.

. Literature review is missing. Based on LR you should define gap you are trying to cover.  NN are powerful tools used in different fields. As a part of that I suggest author read and add below interesting papers:

FEB-Stacking and FEB-DNN Models for Stock Trend Prediction: A Performance Analysis for Pre and Post Covid-19 Periods. Decision Making: Applications in Management and Engineering, 4(1), 51-84.

Model-Based Fuzzy Control Results for Networked Control Systems. Reports in Mechanical Engineering, 1(1), 10-25. https://doi.org/10.31181/rme200101010p

. Validation and comparisons of the results is missing.

. More discussion in on the results of the case study are needed. The authors need to discuss these values and the performance of their approach. How should we know about the quality of these solutions? Could you compare these results with some existing approaches in literature? The improvement must be discussed

. The conclusion section seems to rush to the end. The authors will have to demonstrate the impact and insights of the research. The authors need to clearly provide several solid future research directions. Clearly state your unique research contributions in the conclusion section. Add limitations of the model.  
Scientific soundness :
. The subject addressed in this paper is relevant.  
Interest to the readers :
. In my opinion, the method of this paper seems to be interesting for the readership of the journal. 

Author Response

The authors greatly appreciated your constructive and helpful comments/suggestions. We have included in the attached letter a point-by-point response to the raised concerns. We believe that the manuscript is significantly improved after this revision. Additionally, we have tracked the changes in the revised manuscript in red color.

Reviewer 2 Report

I am very confused by this paper. 

The image processing is very extensive and well thought out but it is needed? For example the Micasense detector is (in our laboratory measurments) barely an 8 bit resolution instrument, and for a drone flying at low altitude, is atmospheric correction necessary? I think not. 

Also, BRDF corrections? What is the surface that is being observed? Is there any notion from previous work that the BRDF correction is required? Most of the fire ash i have measured is Lambertian.

So...you need to consider the accuracy/resolution of the instruments and what corrections are required

I would like to see a trade study of how these various series of corrections affect the output result, if at all. And I would also like to understand how the laboratory measurements have and relevance to the conditions existing post wild- or prescribed-fire.  

Author Response

(The authors gave the same response as above.)

Round 2

Reviewer 1 Report

The authors have addressed the point of my concern. I am happy with their corrections. Hence, I would like to recommend this manuscript to be published.

Reviewer 2 Report

The paper is better now with a few simple corrections. As I stated earlier and as is show in Figure 4 BRDF correction does little on these largely specular targets of interest. 

Overall, a nice paper, well written and organized.